# Comprehensive Insights into the Cholesterol-Mediated Modulation of Membrane Function Through Molecular Dynamics Simulations

**DOI:** 10.3390/membranes15060173

**Published:** 2025-06-08

**Authors:** Ehsaneh Khodadadi, Ehsan Khodadadi, Parth Chaturvedi, Mahmoud Moradi

**Affiliations:** Department of Chemistry and Biochemistry, University of Arkansas, Fayetteville, AR 72701, USA; ekhodada@uark.edu (E.K.); ehsank@uark.edu (E.K.); parthc@uark.edu (P.C.)

**Keywords:** cholesterol, molecular dynamics simulations, liposomes, proteoliposomes, planar bilayers, curved bilayers, lipid ordering, protein–lipid interactions, coarse-grained simulations, all-atom simulations, phase separation, flip-flop dynamics, drug delivery systems

## Abstract

Cholesterol plays an essential role in biological membranes and is crucial for maintaining their stability and functionality. In addition to biological membranes, cholesterol is also used in various synthetic lipid-based structures such as liposomes, proteoliposomes, and nanodiscs. Cholesterol regulates membrane properties by influencing the density of lipids, phase separation into liquid-ordered (Lo) and liquid-disordered (Ld) areas, and stability of protein–membrane interactions. For planar bilayers, cholesterol thickens the membrane, decreases permeability, and brings lipids into well-ordered domains, thereby increasing membrane rigidity by condensing lipid packing, while maintaining lateral lipid mobility in disordered regions to preserve overall membrane fluidity. It modulates membrane curvature in curved bilayers and vesicles, and stabilizes low-curvature regions, which are important for structural integrity. In liposomes, cholesterol facilitates drug encapsulation and release by controlling bilayer flexibility and stability. In nanodiscs, cholesterol enhances structural integrity and protein compatibility, which enables the investigation of protein–lipid interactions under physiological conditions. In proteoliposomes, cholesterol regulates the conformational stability of embedded proteins that have implications for protein–lipid interaction. Developments in molecular dynamics (MD) techniques, from coarse-grained to all-atom simulations, have shown how cholesterol modulates lipid tail ordering, membrane curvature, and flip-flop behavior in response to concentration. Such simulations provide insights into the mechanisms underlying membrane-associated diseases, aiding in the design of efficient drug delivery systems. In this review, we combine results from MD simulations to provide a synoptic explanation of cholesterol’s complex function in regulating membrane behavior. This synthesis combines fundamental biophysical information with practical membrane engineering, underscoring cholesterol’s important role in membrane structure, dynamics, and performance, and paving the way for rational design of stable and functional lipid-based systems to be used in medicine. In this review, we gather evidence from MD simulations to provide an overview of cholesterol’s complex function regulating membrane behavior. This synthesis connects the fundamental biophysical science with practical membrane engineering, which highlights cholesterol’s important role in membrane structure, dynamics, and function and helps us rationally design stable and functional lipid-based systems for therapeutic purposes.

## 1. Introduction

Cholesterol is a critical component of biological membranes, profoundly shaping their structural and functional properties [1]. Because of its amphipathic nature and rigid, planar steroid ring, cholesterol can fit into lipid bilayers and interact with lipid molecules to modify membrane properties such as thickness [2], fluidity [3], permeability [4,5], and stability [6,7,8]. Cholesterol also combines with sphingolipids to form lipid rafts, microdomains where phase separation between liquid-ordered (Lo) and liquid-disordered (Ld) domains occurs. These rafts support membrane integrity and are involved in cell signaling, trafficking, and protein sorting [9,10,11]. These roles extend beyond natural systems to synthetic environments like liposomes and micelles, which are widely used in drug delivery applications [12].

Biological membranes, only a few nanometers thick, serve as dynamic barriers essential for cellular function [13,14,15]. These membranes are composed of a complex mixture of lipid molecules, organized into distinct structural and functional domains regulated by cellular mechanisms [16,17]. Advances in lipidomics and high-resolution mass spectrometry have revealed the chemical diversity of membrane lipids in eukaryotic and bacterial cells [18,19]. However, the mechanisms by which these lipids combine to create functional membranes remain poorly understood [20]. Lipid self-organization and domain formation are critical to membrane properties, supporting protein insertion, signaling, and interactions with the cell environment [21,22]. Changes in lipid composition affect cellular responses to stress, drug resistance, and morphological adaptation [23,24,25]. Deciphering the regulatory factors that maintain membrane integrity and flexibility is crucial for advancing drug delivery and cell signaling technologies [26].

Cholesterol modulates membrane properties in a concentration-dependent manner, with distinct effects across planar [27], curved [28], liposomal [29,30], and proteoliposomal [31] systems. In planar bilayers, cholesterol enhances rigidity, orders lipids, and diminishes permeability [29]. In curved membranes like vesicles, cholesterol increases rigidity and redistributes to regions of lower curvature, stabilizing curvature dynamics and lipid flip-flop behavior [32]. Cholesterol maintains membrane asymmetry by redistributing between bilayer leaflets, balancing mechanical forces [32,33,34,35,36]. In liposomal systems, cholesterol optimizes drug encapsulation and release profiles by modulating bilayer flexibility and packing density [37,38,39]. In proteoliposomes, cholesterol stabilizes membrane proteins and regulates conformational dynamics, contributing to pathological processes like Alzheimer’s disease by accelerating protein aggregation [40].

Molecular dynamics (MD) simulations, both coarse-grained and all-atom, have now become a vital means of examining cholesterol multifunctionality in membranes [41,42,43]. These simulations provide atomistic details on lipid dynamics, phase behavior, and molecular interactions, furthering our knowledge of how cholesterol affects membrane stability, fluidity, and protein dynamics. This review synthesizes recent findings from MD simulations to provide a unified perspective on cholesterol’s regulatory roles in both biological membranes and synthetic systems. Unlike previous reviews that typically focus on a single system or simulation method, our work covers a range of membrane types, including planar bilayers, curved vesicles, liposomes, nanodiscs, and proteoliposomes, and integrates insights from both coarse-grained and all-atom approaches. The objective is to clarify how cholesterol concentration and spatial distribution influence membrane mechanics, organization, and therapeutic potential, with implications for drug delivery and membrane-associated diseases.

## 2. Planar Bilayers

Planar lipid bilayers serve as a fundamental model for investigating the structural and functional dynamics of biological membranes. These membranes are composed of three major lipid classes: glycerophospholipids, sphingolipids, and sterols. Each class contributes uniquely to membrane organization and behavior [44,45]. Hydrophilic head groups are oriented outward toward the aqueous environment, while hydrophobic acyl chains face inward, forming a living, semi-permeable network [4]. Cholesterol, a major sterol in mammalian bilayers, promotes structural stiffness, stability, and resistance to stresses [46]. The precise organization of these lipids, particularly the condensing effect of cholesterol, governs membrane properties such as thickness, fluidity, and permeability. It also facilitates adaptive responses to environmental stimuli, including interactions with therapeutic compounds and solutes [47].

### 2.1. Membrane Thickness and Lipid Ordering

This section provides a detailed discussion of how cholesterol influences membrane thickness and lipid ordering, expanding on the overview provided in the introduction. Cholesterol, a sterol isoprenoid, integrates seamlessly into lipid bilayers due to its amphipathic structure, which consists of a semi-rigid tetracyclic ring and a polar 3β-hydroxyl group [1,48,49]. This unique structure enables cholesterol to condense lipid chains, reducing membrane permeability to non-polar molecules and enhancing the hydrophobic barrier against polar molecules [11,50]. By stabilizing phospholipid interactions, cholesterol promotes the formation of liquid-ordered (Lo) microdomains, which contribute to membrane order and structural integrity [51,52].

This structural activity of cholesterol extends to modulating the shape of planar bilayers, boosting lipid packing density, reducing lateral diffusion and enhancing bilayer thickness. Atomistic molecular dynamics simulations show that cholesterol thickens SOPC lipid membranes significantly: from 3.81 nm in SOPC-free membranes to 4.11 nm in cholesterol-containing membranes at 273 K [53]. This thickening effect persists across a range of temperatures, promoting a more robust and structured membrane. Furthermore, the mass-density profiles of SOPC systems mixed with 10% cholesterol demonstrate distinct density variations along the membrane thickness, emphasizing cholesterol’s role in altering the spatial distribution of membrane components. As shown in Figure 1, the density of lipid tails decreases, while the head group density sharpens near the membrane center. These profiles illustrate how cholesterol contributes to enhanced bilayer packing and ordering, corroborating observations of increased thickness and structural stability at different temperatures.

Studies in SOPC and POPC bilayers reveal that cholesterol induces a liquid-ordered (Lo) phase, maintaining membrane fluidity and preventing phase separation, which stabilizes the bilayer [54]. Cholesterol exhibits strong condensing effects, particularly in bilayers with saturated lipids or specific components such as ceramide (CER) and sphingomyelin (SM), further enhancing membrane order and rigidity [55]. At higher concentrations, cholesterol self-associates into dimers and tetramers, stabilizing specific membrane regions. These aggregates are especially prominent in cholesterol-rich environments, such as viral lipid envelopes, where they influence lipid organization and protein–lipid interactions [56].

Further studies indicate that membrane thickness increases with cholesterol concentration up to 20 mol%, reaching approximately 4.35 nm, before slightly decreasing beyond this point, suggesting a saturation threshold at 20–30 mol%. Similarly, lipid ordering peaks at 30 mol% cholesterol, with SCD values reaching maximum levels before reversing at higher concentrations. These findings emphasize the importance of an optimal cholesterol concentration for maintaining bilayer stability and functionality [54].

In more complex systems involving additional biomolecules, cholesterol’s effects are influenced by interactions with other agents. For example, cholesterol increases bilayer thickness in DPPC membranes, but melatonin counters this effect, particularly in DOPC bilayers, introducing disorder and fluidity into the system. This interplay illustrates cholesterol’s tendency to condense and order the membrane, while melatonin acts as a fluidizing agent, highlighting the regulatory dynamics between cholesterol and other biomolecules [57]. Cholesterol also exerts unique effects in asymmetric bilayers. Increased cholesterol concentrations enhance bilayer thickness, lipid tail order, and stability while reducing lateral lipid diffusion and cholesterol flip-flop between leaflets. These effects help maintain asymmetry and stability in biological membranes. Comparisons between cholesterol and ergosterol, a sterol prevalent in fungal membranes, reveal that cholesterol’s planar structure has stronger condensing effects, underscoring its unique role in mammalian membranes [58,59].

As demonstrated in Figure 2, the temperature dependence of the average tilt angle of DPPC molecules with respect to the membrane normal indicates that cholesterol reduces the overall tilt of DPPC molecules more effectively than ergosterol. This reduction in tilt highlights cholesterol’s ability to induce tighter packing and structural order in the bilayer. The inset of the figure further provides a visual representation of the DPPC tilt vector at 290 K, schematically emphasizing sterol-induced alignment. Such alignment not only stabilizes the membrane but also impacts the ordering dynamics, where cholesterol’s enhanced planarity contributes to its stronger ordering effects compared to ergosterol. This structural advantage enables cholesterol to reinforce membrane integrity under varying thermal conditions.

### 2.2. Mechanical Properties and Membrane Rigidity

The presence of cholesterol significantly enhances the mechanical stability of lipid bilayers by increasing their bending rigidity and elasticity. Experimental data, such as small-angle neutron scattering (SANS) and solid-state ^2^H NMR spectroscopy in Figure 3, demonstrate that cholesterol reduces the lipid area and enhances packing density in unsaturated DOPC bilayers. These changes lead to a threefold increase in bending rigidity as concentrations approach 50%. This stiffening effect is critical in biological processes like viral budding and protein–lipid interactions [60]. Computational advancements, such as local thickness fluctuation (LTF) analysis, which measures membrane deformation under thermal fluctuations, further confirm cholesterol’s role in improving membrane stability and elasticity [61].

In complex bilayers, such as axonal membranes enriched with sphingomyelin (SM) or galactosylceramide (GalCer), cholesterol strengthens lipid–lipid interactions, increasing bilayer stiffness and membrane resilience. These properties are particularly important in myelin sheaths, where cholesterol contributes to mechanical resistance and protects against deformation and water penetration [62].

### 2.3. Asymmetry and Leaflet Coupling

In planar bilayers with an asymmetric distribution, cholesterol moves between leaflets to offset structural biases introduced by a difference in lipid composition or ion gradients. This kinetic redistribution stabilizes important properties like lipid area and lateral pressure distributions, stabilizing bilayers and leaflet asymmetry that regulates proper membrane function [63]. Figure 4, which shows the order parameters of sn-2 lipid tails at various cholesterol concentrations, presents how cholesterol alters lipid packing asymmetry and leaflet coupling in DPPC membranes. These findings highlight cholesterol’s ability to extend its stabilizing and ordering effects to asymmetric bilayers by mitigating leaflet imbalances and reinforcing membrane structure. Also, cholesterol’s rapid diffusion between bilayer leaflets is efficient in alleviating mechanical stresses, hence its importance in keeping systems balanced and compact in asymmetric environments [56].

### 2.4. Phase Separation and Fluidity

Cholesterol interacts with phospholipids to form liquid-ordered (Lo) microdomains, which serve as platforms for molecular organization and phase separation [52,64]. Coarse-grained MD simulations reveal key insights into cholesterol’s spatial arrangement and its critical role in balancing phase separation and fluidity within lipid bilayers. The density distribution functions, as shown in Figure 5, illustrate cholesterol’s preferential localization within bilayer leaflets and the bilayer center, depending on concentration. At higher cholesterol concentrations ($x_{\mathrm{chol}}=0.5$), an additional density peak at the bilayer center highlights cholesterol’s stabilizing effect on liquid-ordered domains. The influence of lipid unsaturation further modulates cholesterol’s solubility and spatial distribution, affecting its ability to maintain structural organization and membrane dynamics [65,66].

This diffusion behavior at the bilayer center contributes to overall membrane fluidity by influencing how lipids are organized and packed within these domains [67]. Additionally, the condensing effect of cholesterol, especially pronounced in polyunsaturated bilayers, reduces the area per lipid, promoting tighter packing. This condensing effect is further evidenced by the free energy profile in panel (D) of Figure 5, which shows cholesterol’s metastable state at the bilayer center, indicating its strong association with lipid molecules at higher concentrations. The free energy barrier ($\Delta F^{‡}$) for cholesterol movement, shown in panel (E), underscores how higher concentrations of cholesterol reduce its mobility between the bilayer center and leaflets, contributing to the structural stability of liquid-ordered regions.

When cholesterol is present in balanced proportions with lipids, electron density profiles reveal its close association with lipid molecules, enhancing phase separation and contributing to structural stability within ordered regions. This ordering effect is particularly prominent in bilayers with long-tailed, saturated lipids, where cholesterol’s planar structure facilitates faster formation of liquid-ordered (Lo) phases. Simulation snapshots, as depicted in panel (C) of Figure 5, further demonstrate cholesterol’s preferential arrangement, with molecules in the bilayer center exhibiting faster diffusion dynamics compared to those within the leaflets. Such phase separation stabilizes the membrane and modulates fluidity, especially in unsaturated lipid-rich systems, reinforcing cholesterol’s role in regulating membrane fluidity and stability across different lipid compositions [68].

Further examples of cholesterol’s modulation of membrane dynamics come from research on its derivatives and interactions with the environment. Chromosome analogs of cholesterol (CHIMs) replicate its membrane-ordering properties even at low levels and are therefore an excellent resource for understanding cholesterol’s impact on membrane behavior and as tools to study its function under a wide variety of conditions [69]. Similarly, experimental studies have shown that ibuprofen disrupts the molecular ordering of cholesterol-loaded POPC bilayers, increasing molecular mobility without significantly altering membrane permeability [70].

Cholesterol’s influence reaches beyond its multiple roles in membrane organization, which include regulating structure and thickness, as well as fluidity and phase behavior. Cholesterol’s concentration-dependent effects reveal a ’sweet spot’ where structural integrity and membrane function are optimized, even under stress conditions. Furthermore, cholesterol stabilizes lipid rafts—microdomains dense with cholesterol and sphingolipids. These rafts of lipids form Lo (liquid-ordered) domains that serve as bases for cell signaling and protein clustering, the basis of so much cell activity. Cholesterol’s distribution within these regions enhances the thickness and rigidity of the membrane, particularly in sphingolipid-rich regions, facilitating the formation and repair of lipid rafts and driving signals, protein folding, and membrane trafficking [68].

The impact of cholesterol on phase separation is also lipid saturation-dependent. In saturated lipid bilayers such as DPPC or DSPC, cholesterol integrates more effectively, promoting the formation of tightly packed, liquid-ordered (Lo) domains. In contrast, in polyunsaturated lipid bilayers like DOPC or DLiPC, cholesterol’s ability to induce ordering is reduced due to greater chain disorder and increased spacing between lipid tails. This variation influences membrane rigidity and the extent of phase separation, leading to more dynamic and less ordered domain formation in unsaturated systems [71,72,73].

In addition to cholesterol, other sterols such as ergosterol and phytosterols have been studied for their effects on membrane structure and dynamics. Ergosterol, the principal sterol in fungal membranes, shares structural similarities with cholesterol but exhibits weaker condensing effects and reduced ordering capabilities. Simulation studies and experimental data show that cholesterol reduces the tilt angle of saturated lipids more effectively than ergosterol, enhancing membrane order and thickness. These differences are attributed to cholesterol’s more planar structure and stronger van der Waals interactions with lipid tails, which make it more efficient in stabilizing lipid domains [74,75]. Including these comparative perspectives further emphasizes cholesterol’s distinctive regulatory role in membrane behavior.

## 3. Curved Bilayers

Cholesterol modulates membrane properties, especially in curved bilayers like vesicles or spherical membranes. Its distribution and dynamics in these environments are central to understanding processes such as vesicle formation, membrane fusion, and protein sorting. MD simulations have played a central role in understanding the role of cholesterol in modulating membrane curvature, stability and flip-flop properties across bilayer leaflets.

It is well known that membrane curvature is key to biological functions, including cholesterol’s function in curved bilayers. Molecular dynamics simulations revealed that cholesterol modulates membrane structure at different curvatures, influencing important cellular processes like vesicle assembly and protein sorting. In curves, cholesterol regulates membrane function, which is fundamental in dynamic membrane change. Cholesterol also acts as a stabilizing agent during stress such as lipid oxidation and electroporation, where rapid pulses of electric current produce minute pores. By countering oxidative damage and stabilizing dynamic pores, cholesterol helps preserve membrane integrity under such conditions [76,77].

In regions with extreme curvature, such as fusion pore necks, cholesterol is often excluded from highly thinned areas, facilitating pore constriction and collapse. This behavior underscores cholesterol’s preference for more rigid, less curved membrane regions, highlighting its stabilizing influence in fusion dynamics. Furthermore, cholesterol stabilizes deformed membranes by interacting with transmembrane proteins, such as CD81, preventing significant curvature-induced deformations. These interactions extend cholesterol’s stabilizing effects beyond the lipid bilayer to include membrane-associated proteins, emphasizing its dual role in maintaining membrane architecture and supporting protein dynamics [78,79].

### 3.1. Cholesterol-Induced Stability

Stabilizing actions of cholesterol have been extensively studied and showed to contribute to better membrane order and stability. It has been demonstrated that cholesterol adsorbs local membrane order within lipid bilayers to enhance the adsorption of PAMAM dendrimers into cholesterol-laden solutions. These results reinforce cholesterol’s membrane-structure preservation, and raise the possibility of its application to drug delivery, in which membrane integrity is an essential consideration [80]. Cholesterol’s impact on the partitioning of drugs, such as amantadine (AMT), into lipid bilayers has also been explored. By increasing the order and rigidity of lipid tails, cholesterol reduces AMT partitioning, limiting the membrane’s capacity to accommodate the drug. This behavior aligns with findings on cholesterol’s broader influence on membrane permeability and drug interactions, particularly within cholesterol-rich viral membranes [81].

Cholesterol distribution across the inner and outer monolayers of curved membranes varies significantly with curvature. In highly curved regions of the inner monolayer, cholesterol molecules are positioned closer to the bilayer center, while in the outer monolayer, they are more evenly distributed. This spatial reorganization mitigates curvature-induced stress, stabilizing the membrane and maintaining its structural integrity. Figure 6 provides visual evidence of these phenomena. The left panel shows how curvature influences the spatial distribution of cholesterol molecules, with the inset highlighting the peak distances of cholesterol’s hydroxyl groups from the membrane surface. The right panel complements this by illustrating snapshots of membrane configurations under varying curvature, highlighting the dynamic adaptations of cholesterol molecules and lipid species.

The ability of cholesterol to adapt dynamically to different curvatures is crucial for stabilizing membranes under stress. For instance, cholesterol enhances membrane resistance to electroporation, reducing the likelihood of breakdown when exposed to high electric fields [82]. This effect is particularly pronounced in cholesterol-rich membranes, where its ordering properties counteract disruptions caused by external forces. Furthermore, cholesterol regulates membrane contacts with other molecules, like amphiphilic nanoparticles, by increasing the rigidity and packing density of lipid bilayers. Thus, cholesterol increases the energy barrier of nanoparticle penetration, thus demonstrating its shielding action to preserve membrane integrity [83]. These results show that cholesterol is fundamental to membrane stability and pliability in physical and chemical environments.

### 3.2. Flip-Flop Dynamics

Cholesterol’s flip-flop behavior between bilayer leaflets is a key mechanism in maintaining membrane asymmetry. Studies using MD simulations have shown that cholesterol diffuses rapidly at the bilayer center, contributing to dynamic heterogeneity and asymmetry between the leaflets. This asymmetry arises from differences in composition between the upper and lower leaflets, with cholesterol often distributing unevenly due to its movement between them. Cholesterol’s tendency to reside preferentially in one leaflet or at the bilayer center results in compositional differences that influence each leaflet’s properties and contribute to overall membrane stability. These observations align with findings on cholesterol’s behavior in ordered membrane regions, demonstrating how its dynamic movement supports membrane structure [67].

In polyunsaturated bilayers, cholesterol frequently adopts a flipped orientation, with its hydroxyl group pointing toward the membrane center. This behavior is facilitated by the increased disorder of polyunsaturated environments, highlighting how cholesterol’s mobility is influenced by the surrounding lipid composition. This orientation flexibility provides additional insight into how cholesterol’s movement between leaflets enhances membrane flexibility and maintains asymmetry [68].

Additionally, recent studies have shown that cholesterol’s redistribution across curved membranes is closely linked to local curvature dynamics. Cholesterol exhibits a strong preference for negatively curved regions, where its asymmetry between leaflets can exceed 80%, driven by its negative spontaneous curvature. This redistribution alleviates mechanical stress and supports the structural organization of lipid membranes, particularly in di-unsaturated bilayers like DOPC. The coupling between cholesterol’s movement and curvature further underscores its role in stabilizing membranes under dynamic conditions [84].

Figure 7 visually illustrates the coupling between membrane curvature and cholesterol asymmetry in DPPC and DOPC bilayers. As seen in the time-dependent curvature analysis (panels a–d), cholesterol’s presence significantly alters the distribution of mean curvature, particularly in di-unsaturated systems like DOPC. Panel e highlights the asymmetry of cholesterol between the leaflets, which increases in regions with high curvature. The linear relationship between time-averaged mean curvature and cholesterol asymmetry (panel f) further emphasizes how curvature dynamics influence cholesterol distribution, revealing its critical role in stabilizing membranes with diverse curvature profiles.

More detailed investigations of cholesterol transport in asymmetric lipid bilayers indicate that cholesterol is more inclined to flow to leaflets with more order to reduce mechanical strain. This selective migration strengthens cholesterol’s capacity to moderate mechanical stresses in the bilayer due to the membrane’s internal organization. Together, these results offer a comprehensive picture of cholesterol’s flip-flop mechanisms in preserving membrane shape under diverse environmental conditions [65]. In addition, cholesterol typically results in membrane stiffness that raises the energy level for transverse diffusion or flip-flops. It also affects order and stiffness, especially in asymmetric bilayers, causing lower rates of spontaneous lipid flip-flop because of the structural stability it confers. This greater rigidity is problematic for mimicking precise flip-flop dynamics, because it affects the ease with which cholesterol and other lipids can transition between leaflets [43].

## 4. Nanodisc

Cholesterol is a fundamental regulator of nanodisc stability and behavior in both natural and artificial systems [85]. Since high-density lipoprotein (HDL) nanodisc cholesterol plays a role in reverse cholesterol transport (RCT) [86], apolipoprotein A-I (APOA1) binds to cells for cholesterol efflux and transport [87]. APOA1’s flexibility also enables stable HDL nanodiscs of discoidal and spherical shapes and size and shape of HDL particles [88]. Synthetic nanodiscs stabilized with membrane scaffold proteins (MSPs) or APOA1-like peptides contain cholesterol, which maintains stability and is compatible with membrane proteins, an ideal physiological system for understanding protein–lipid interactions [89,90]. These cholesterol-anchored nanodiscs are useful as tools in biophysical research, revealing structural adaptation and protein–lipid dynamics [91,92].

### 4.1. Role of Cholesterol in Reverse Cholesterol Transport (RCT)

Cholesterol plays a role in reverse cholesterol transport (RCT) [93], which works by interacting with HDL nanodiscs [94]. RCT starts when APOA1 (apolipoprotein A1) encounters the ATP-binding cassette transporter ABCA1 on the cell surface, and spills cholesterol out of cells like macrophages. This process produces developing, discoidal HDL (nHDL) stabilized by APOA1 [95]. Next, lecithin acyltransferase (LCAT) turns free cholesterol into cholesteryl esters and produces mature, spherical HDL (sHDL) [96]. More recently, cholesterol efflux capacity has been shown to play a role in RCT, and the interaction of APOA1 with ABSA1 stabilizes HDL nanodiscs while moving cholesterol [97,98,99,100]. Also important for cardiovascular health is the ability of HDL to transfer cholesterol; LCAT helps to make discoidal HDL into spherical particles [101,102].

### 4.2. Structural Adaptation and Stability in Synthetic Nanodiscs

The synthetic nanodiscs are patches of lipid bilayer held together by membrane scaffold proteins (MSPs) or APOA1-like peptides, which afford a static environment for cholesterol’s effects on membrane proteins [103]. Cholesterol helps stabilize nanodiscs by encapsulating natural lipid milieus, helping to explore protein–lipid dynamics without detergents [89,91]. Cholesterol also alters protein arrangements and lateral pressure curves in the bilayer, revealing a more physiologically sound model of membrane dynamics [104].

### 4.3. Impact of Cholesterol on Protein Configuration and Nanodisc Shape

Cholesterol also alters protein sequences in nanodiscs, influencing APOA1’s structural flexibility [88]. APOA1, for example, can form a double-belt or picket-fence shape around the nanodisc, and cholesterol facilitates this flexibility [105]. This cholesterol-induced structural flexibility increases cholesterol efflux and maintains nanodisc vesicles throughout RCT, contributing to their capability for transporting cholesterol [106].

Figure 8 presents lecithin–cholesterol acyltransferase (LCAT) dynamics in nanodiscs. It shows how LCAT evolved structurally, bending toward the nanodisc edge during simulation time. They demonstrate how cholesterol and its kinases such as LCAT respond to curvature and surface of nanodiscs, altering how nanodiscs organize and function in reverse cholesterol transport. In addition, the alignment between LCAT’s spatial position and the nanodisc surface geometry further highlights cholesterol’s role in stabilizing such structures.

## 5. Liposome

Cholesterol plays a key modulatory role in shaping the structure and function of liposomes by influencing bilayer flexibility, packing, and stability. Its density within the bilayer can affect self-formation, stability, encapsulation, and release of drugs, and therefore plays a critical role in liposome-based drug delivery systems. Numerous studies utilizing molecular dynamics simulations have contributed to understanding how cholesterol modulates these properties in various lipid environments [107,108,109].

### 5.1. Formation and Stability

Cholesterol’s influence on bilayer properties was further elucidated through molecular dynamics simulations. The simulations demonstrated that increasing cholesterol concentration (0 to 70 mol%) expanded the bilayer, as evidenced by an increase in the area per molecule and a concurrent decrease in bilayer thickness. This behavior differs from the condensation phenomenon observed in normal lipid bilayers like DOPC or DMPC. Interestingly, optimal flexibility and stability were obtained with cholesterol levels ranging from 40 to 50 mol%, which is indicated by a high isothermal area compressibility. They then visualized those observations in simulations, with the niosome bilayers’ structural changes at different cholesterol levels, as shown in Figure 9. This information also illustrates the importance of cholesterol to regulate bilayer behavior, moving from a gel to a liquid-ordered phase, and maintaining stability through enhanced hydrogen bonding in the Span60–cholesterol–water complex. This flexibility makes it valuable for defining powerful drug delivery systems [2].

Other coarse-grained molecular dynamics simulations drew further on these observations, and examined systems that mixed several phospholipids (DLPE, DOPE, DLiPE) with cholesterol and vitamin C; they showed that cholesterol levels ranging from 15% to 40% enhanced liposome production, as cholesterol migrated into the bilayer and vitamin C sealed in the aqueous center. As cholesterol concentration increased, liposome stability improved, evidenced by more compact bilayer structures and extended lifetimes. This research highlights cholesterol’s role in enhancing phospholipid packing density and increasing membrane thickness, which is essential for stabilizing liposomes, particularly when encapsulating agents like vitamin C [38].

Expanding the understanding of cholesterol’s role, additional studies investigated its impact on ultrasound-responsive liposomes with varied lipid compositions (DSPC, DSPE-PEG, DOPE, MSPC) and cholesterol concentrations from 0% to 40%. Findings revealed that cholesterol restricted lipid diffusion, which increased bilayer rigidity and helped stabilize the membrane under ultrasound conditions. This stabilization can modulate drug release efficiency by limiting membrane permeability. In comparison, lipid types such as DOPE, which promote higher curvature and lipid mobility, were associated with increased drug release. Such results highlight the role of cholesterol and lipid types like MSPC in influencing liposome behavior for best-fit drug delivery systems [110].

Cyclodextrins (CDs), particularly methyl-*β*-cyclodextrin (M*β*CD), are widely used in both experimental and therapeutic contexts to extract or deliver cholesterol to membranes, thereby modifying membrane composition and key properties such as rigidity, permeability, and stability. These cyclic oligosaccharides form inclusion complexes with cholesterol and are commonly employed to manipulate membrane cholesterol levels and support drug delivery strategies. Molecular dynamics simulations have provided atomistic insights into the mechanism of cholesterol extraction by β-CDs. Specifically, it has been shown that β-CD dimers, when oriented perpendicular to the bilayer surface, can spontaneously extract cholesterol molecules, forming a stable 2:1 CD:cholesterol complex. This process is energetically favorable and involves local disruption of lipid packing, which facilitates the insertion and retention of cholesterol within the cyclodextrin cavity. These findings highlight the dual role of CDs as cholesterol regulators and potential nanocarriers in biomedical applications [111,112].

### 5.2. Drug Loading and Release

Along with its structural effects, cholesterol is important in controlling the loading and release of drugs from liposomes [113]. Studies have shown that cholesterol can modulate the encapsulation efficiency of drugs and control release kinetics by altering the membrane’s rigidity and lipid packing [29,30,114]. This is evident from differential scanning calorimetry (DSC) analyses, where cholesterol and elastin-like recombinamer (ELRCTA) intercalation into DPPC liposomes reduced the phase transition temperature and increased membrane fluidity [115]. As shown in Figure 10, these changes highlight cholesterol’s role in disrupting van der Waals forces within the bilayer and enhancing fluid-like behavior, particularly at higher concentrations. The schematics further illustrate the molecular-level interactions that modulate bilayer properties and encapsulation efficiency.

The balance between cholesterol concentration and lipid composition determines how tightly drugs are retained within the liposome and how effectively they are released, especially under specific stimuli such as ultrasound. Cholesterol’s ability to promote rigidity in the membrane reduces the exposure of charged head groups, thereby increasing the shelf life of encapsulated drugs by preventing premature leakage. This rigidity, which is concentration-dependent, also reduces the likelihood of micelle fusion with each other and with cell membranes, enhancing the stability of the liposomal system during storage and upon administration.

### 5.3. Phase Behavior and Structural Organization

The phase behavior of cholesterol-containing bilayers and micelles underscores cholesterol’s considerable influence on membrane dynamics. Cholesterol promotes the formation of liquid-ordered (Lo) phases within liposomal bilayers, enhancing both membrane rigidity and stability. This condensing effect facilitates a transition from a gel phase to a liquid-ordered phase, particularly at intermediate concentrations (15–40%), contributing to the coexistence of distinct Lo and liquid-disordered (Ld) domains. These domains are not merely structural features but play crucial roles in defining the functional behavior of liposomal systems [38].

Cholesterol-containing bilayers play a critical role in molecular partitioning. The COSMOmic approach, as depicted in Figure 11 (Top), illustrates the partitioning behavior of solutes across bilayers, showcasing the organization of lipid head groups (red), lipid tailgroups (yellow), and water (blue). These structural features influence free energy profiles and the propensity of solutes to locate within specific bilayer regions. Cholesterol’s presence modifies the bilayer’s polarity gradient, creating a distinct region favorable for solute accumulation near the bilayer’s hydrophobic core [116].

The number density profiles in Figure 11 (Bottom) reveal differences between pure DPPC bilayers and those with 30% cholesterol. The addition of cholesterol increases the bilayer’s thickness and reduces the flexibility of DPPC molecules, as evidenced by a narrower solvent-accessible surface area (SASA) distribution for DPPC conformers in cholesterol-containing systems. These structural changes stabilize the Lo phase while maintaining regions conducive to dynamic processes such as pore formation, which is crucial for controlled drug release [110].

These findings highlight cholesterol’s role as a key determinant in liposome behavior, influencing bilayer formation, stability, and drug release. By fine-tuning cholesterol concentrations, it is possible to design liposomal systems with optimized structural properties and tailored therapeutic capabilities. The ability to predict and analyze free energy profiles using methods like COSMOmic provides valuable insights into phase behavior and structural organization in complex lipid systems [116].

## 6. Proteoliposomes

Cholesterol has significant effects on the stability, structure, and conformational behavior of membrane proteins in proteoliposome [60]. It modulates membrane protein interactions with lipid bilayers, with cholesterol concentration modulating protein stability, binding, and aggregation [117].

### 6.1. Membrane Protein Stability

Cholesterol’s interactions with membrane proteins play an important role in maintaining protein function in proteoliposomes. Molecular dynamics simulations revealed that higher cholesterol levels (0%, 20% and 40%) in POPC lipid bilayers cause significant structural changes such as thicker bilayers and more structured lipid packing, as shown in Figure 12 (Top). These cholesterol-induced changes promote hIAPP’s adsorption and attachment to the bilayer through enhanced electrostatic interactions between hIAPP’s C-terminal residues and lipid head groups. In addition, cholesterol is used to build ionic bridges with calcium ions, which help maintain protein–membrane interactions. The adsorption or desorption of hIAPP on bilayers with varying cholesterol content, depicted in Figure 12 (Bottom), highlights the role of cholesterol in modulating these interactions.

Cholesterol also modulates interactions with other peptides and proteins, such as melittin. In POPC bilayers, cholesterol increases lipid packing density and surface hydrophobicity, enhancing hIAPP adsorption and stabilizing its β-sheet structure, which may contribute to cytotoxicity in type II diabetes [118]. Similarly, cholesterol redistributes in response to melittin, disrupting lipid organization and stabilizing the bilayer against peptide-induced damage. These interactions underscore cholesterol’s critical role in peptide-mediated membrane remodeling and its broader influence on protein–lipid dynamics [119].

### 6.2. Structural Changes in Proteoliposomes

Cholesterol not only affects protein stability but also induces significant structural changes in proteoliposomes, especially during interactions with external proteins. Studies using molecular dynamics simulations have shown that cholesterol plays a crucial role in the behavior of listeriolysin O (LLO), a cholesterol-dependent cytolysin, within DOPC-cholesterol bilayers. Cholesterol accumulates near specific recognition motifs in the D4 subunits of LLO, stabilizing both the membrane-bound (MB) and membrane-inserted (MI) states. This accumulation results in a reduction in lipid mobility in the extracellular leaflet, while the MI state induces lipid reorientation, leading to the formation of a toroidal lipid structure around the pore. This reorganization of lipids and cholesterol underscores cholesterol’s essential role in stabilizing the pore-forming configuration, providing insights into how cholesterol modulates the dynamic properties of lipid bilayers during bacterial pore formation [120].

### 6.3. Conformational Dynamics

Cholesterol has profound effects on protein conformation in proteoliposomes, including neurodegenerative disorders. It turns out that cholesterol also regulates amyloid-β (Aβ) aggregation, modulating its shape and engagement with membrane surfaces. Time-lapse atomic force microscopy and molecular dynamics simulations reveal that cholesterol accelerates Aβ aggregation, enlarging aggregate size and enhancing dissociation into the bulk solution. These effects suggest that cholesterol facilitates Aβ-aggregating molecules and might even generate neurotoxic oligomers during Alzheimer’s disease [121,122].

Beyond Aβ, cholesterol also affects other membrane proteins, including G protein-coupled receptors (GPCRs). Simulations at the molecular dynamics scale demonstrate how cholesterol binds CB1 (a Class A GPCR) into its idle state through an electrostatic and hydrogen bond network. Special interactions like the D163-S390 hydrogen bond in TM7 hold the receptor in the active state, and the inactive form is more structurally heterogeneous because these interactions are not maintained. These dynamic switches are what make CB1 necessary for signal transduction and receptor activation, and cholesterol’s effects on protein chemistry and membrane dynamics are key [123].

## 7. Conclusions

The critical effect of cholesterol on membrane properties has been well understood by MD simulations, and a detailed analysis shows its complex effects across a wide range of membranes [1,2]. From planar bilayers to curved vesicles to proteoliposomes, cholesterol’s effect on membrane stability, lipid order, phase, and protein interactions opens a window into the mechanisms by which cholesterol shapes and behaves [28,29]. By regulating liquid-ordered and liquid-disordered domains in planar bilayers, cholesterol drives membrane thickness and lipid ordering, and regulates phase stability. Such effects are critical for membrane stability and function.

In curved bilayers and vesicles, cholesterol modulates curvature dynamics and stabilizes lower curvature regions, which are essential for vesicle growth and membrane fusion. Its constant redistribution among leaflets emphasizes how it helps to maintain bilayer asymmetry and mechanical equilibrium. For liposomes, cholesterol aids in the encapsulation and release of drugs by regulating bilayer flexibility and packing density. Its concentration-dependent phase behavior and structure make it vital for the design of effective drug delivery systems. The same goes for proteoliposomes, where cholesterol locks membrane proteins, governs their conformational changes, and drives peptide aggregation, with consequences for Alzheimer’s disease.

In addition to these functional roles, cholesterol’s transient redistribution across bilayer leaflets emphasizes its central role in maintaining bilayer asymmetry and mechanical stability that are critical for cell and synthetic life. The combined coarse-grained and all-atom MD simulations have yielded invaluable atomic insights, bridging basic biophysics with practical membrane engineering. This commentary brings together these insights and highlights cholesterol’s versatility across membrane systems.

Despite these advances, several important questions remain unanswered. Future work should explore how cholesterol behaves under physiologically relevant stresses, such as changes in pH, oxidative stress, ionic strength, and mechanical deformation. These conditions more closely mimic biological environments and may uncover new roles for cholesterol in health and disease. Further research to develop this strand would need to concentrate on understanding the role of cholesterol in heterogeneous and asymmetric bilayer systems in physiologically plausible conditions. Moreover, studies that focus on cholesterol’s interactions with other sterols (ergosterol, phytosterols) can reveal competitive or synergistic effects that further modify membrane function.

Incorporating hybrid computational methods such as quantum/molecular mechanics (QM/MM) and enhanced sampling techniques (e.g., umbrella sampling, metadynamics) could help resolve rare events like lipid flip-flop, phase separation, and membrane remodeling. Such approaches hold tremendous promise for describing how cholesterol–lipid and cholesterol–protein interactions operate at the atomic level. Additionally, combining machine learning and artificial intelligence (AI) with molecular simulations can accelerate data analysis and improve predictive models of cholesterol behavior across different membrane systems.

Experimental validation remains crucial to complement computational findings. The newer techniques, including single-molecule force spectroscopy and neutron reflectometry, could give quantitative evidence for cholesterol’s role in membrane mechanics, enhancing computational models and bringing theory and practice closer together. This is a field where collaborative work between experimentalists and computational scientists will be the key to progress.

From an application perspective, future studies should investigate the design of cholesterol-enriched lipid systems tailored for specific biomedical applications. By optimizing cholesterol in liposomal drug delivery, we can enhance drug encapsulation rates, release characteristics, and targeting ability. Likewise, understanding cholesterol’s function in preventing the collapse of proteoliposomes might give us more effective biomimetic systems for investigating protein–lipid interactions and drug delivery systems.

The role of cholesterol in disease pathophysiology, including the impact on membrane function in neurodegenerative diseases, cancer, and cardiovascular conditions, also deserves further attention. Knowing cholesterol’s role in pathological processes, such as amyloid aggregation or tumor spread, might help to shape targeted treatments. More significantly, cholesterol’s promise as a drug target for the suppression of membrane-dependent signaling pathways creates exciting opportunities for drug discovery.

Expanding cholesterol’s diverse functions through more efficient computational modeling, experimental validation, and cross-disciplinary work will deepen our understanding of membrane biophysics and open up new opportunities for therapeutic and technological development. These projects will continue to push the frontiers of membrane science with potentially dramatic implications for drug delivery, biomaterials, and disease modeling.

## Figures and Tables

**Figure 1 membranes-15-00173-f001:**
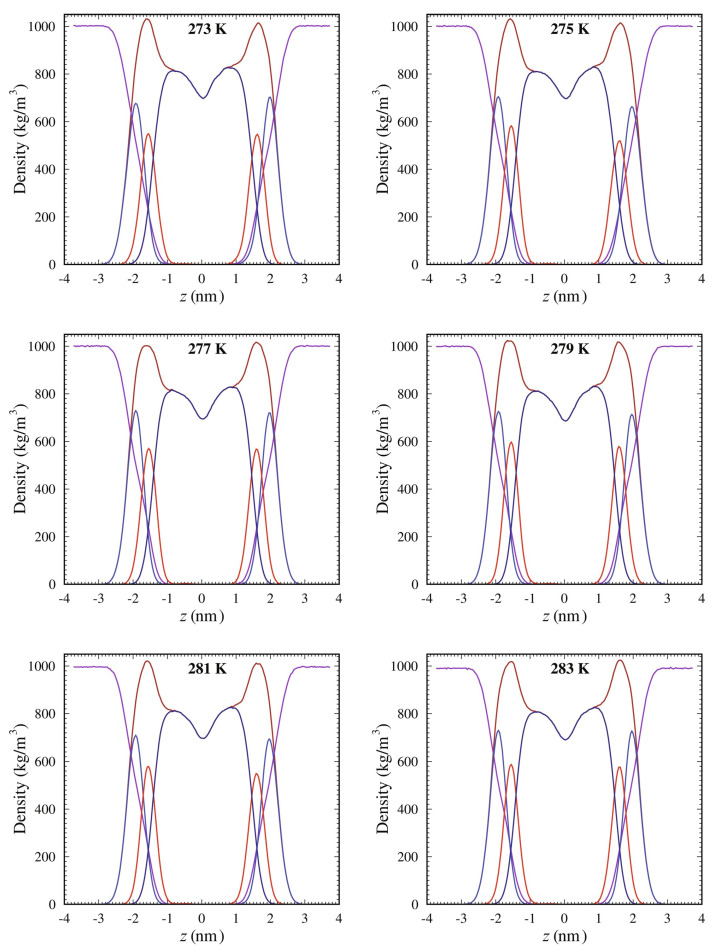
Mass−density profiles of SOPC systems mixed with 10% cholesterol, showing density variations along the membrane thickness (*z*-axis, i.e., the axis perpendicular to the bilayer plane) at different temperatures. The components are represented as follows: lipid (brown), water (dark violet), head groups (steel blue), glycerol ester (red), and tails (dark blue). The center of the membrane is located at z=0. Reproduced with permission from [53]. Copyright (2021) Elsevier.

**Figure 2 membranes-15-00173-f002:**
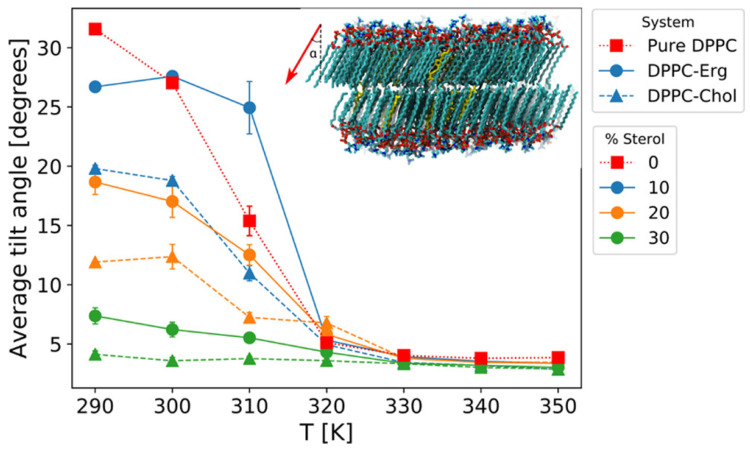
Temperature dependence of the average tilt angle of DPPC molecules with respect to the membrane normal for pure DPPC bilayers and binary systems containing 10% cholesterol or ergosterol. The tilt angle is calculated as the angle (α) between the vector connecting the first and last carbon atoms of each DPPC chain and the z-axis. The inset illustrates a bilayer snapshot at 290 K, showing the upper leaflet’s average DPPC tilt vector (red arrow) relative to the z-axis, highlighting sterol-induced structural order. Reproduced with permission from [59]. Copyright (2021) American Chemical Society.

**Figure 3 membranes-15-00173-f003:**
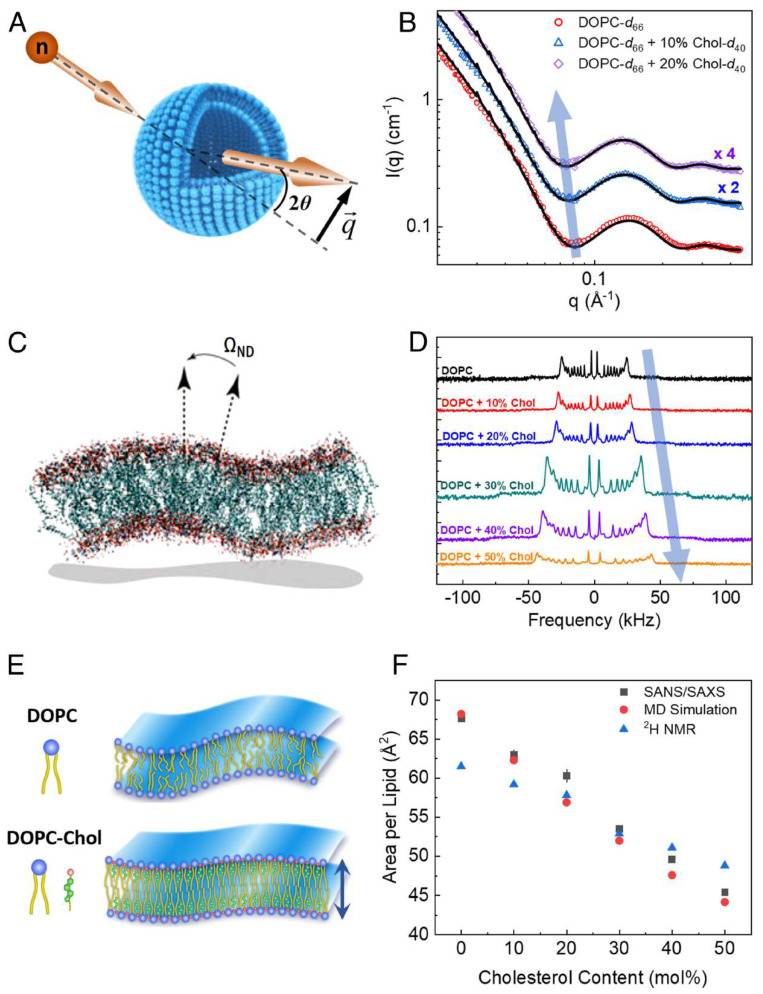
Cholesterol−induced structural changes in lipid membranes. (**A**) Schematic of neutron scattering from lipid vesicles, showing scattering angle 2θ and wavevector transfer. (**B**) SANS data reveal membrane thickening in tail-perdeuterated DOPC-Chol vesicles with increasing Chol content, as seen in the decreasing *q* values of the scattering intensity minimum (lines represent fits to the data). (**C**) Simulation snapshot shows membrane fluctuations and the geometry of director vectors used in ^2^H NMR analysis. (**D**) ^2^H NMR spectra demonstrate increased acyl-chain ordering in multilamellar DOPC dispersions, indicated by rising quadrupolar splitting of the POPC-d_31_ probe with increasing Chol content. (**E**) Diagram illustrates how Chol increases membrane thickness and lipid packing, as supported by (**F**), which shows decreasing average area per lipid with increasing Chol content based on SANS/SAXS data, ^2^H NMR, and RSF-MD simulations. Error bars represent ±1 SD and may be smaller than the symbols. Reproduced from [60], with permission. Copyright (2020) National Academy of Sciences.

**Figure 4 membranes-15-00173-f004:**
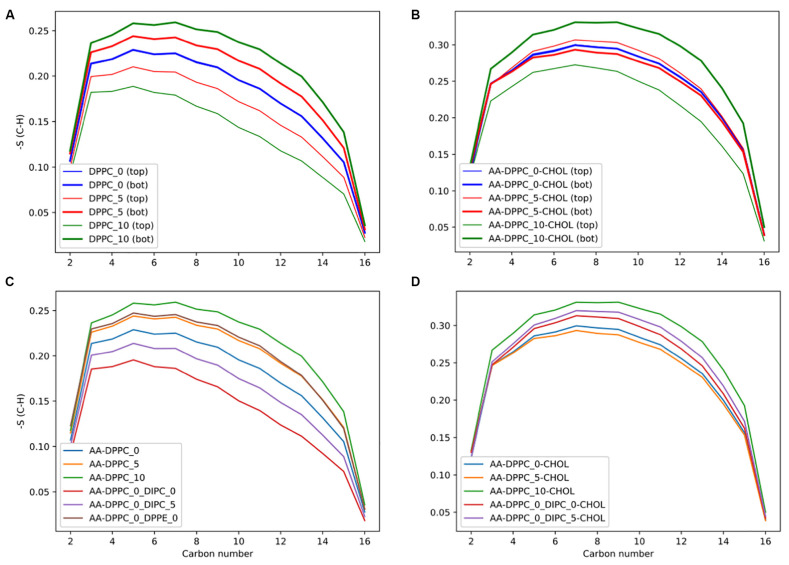
Order parameters of sn-2 lipid tails in asymmetric bilayers: the impact of cholesterol and lipid ratios. (**A**) Coarse-grained DPPC bilayers with 0%, 5%, and 10% CHOL, comparing top and bottom leaflets. (**B**) All-atom DPPC bilayers with the same cholesterol concentrations, showing leaflet-specific differences. (**C**) All-atom DPPC systems mixed with DIPC or DPPE at different ratios and 0% CHOL. (**D**) All-atom DPPC:DIPC mixtures with increasing CHOL concentration (0%, 5%, 10%). The plots show how lipid ordering varies between top and bottom leaflets of an asymmetric membrane. Reused from [63], licensed under Creative Commons Attribution 4.0 International (CC BY 4.0) (https://creativecommons.org/licenses/by/4.0/).

**Figure 5 membranes-15-00173-f005:**
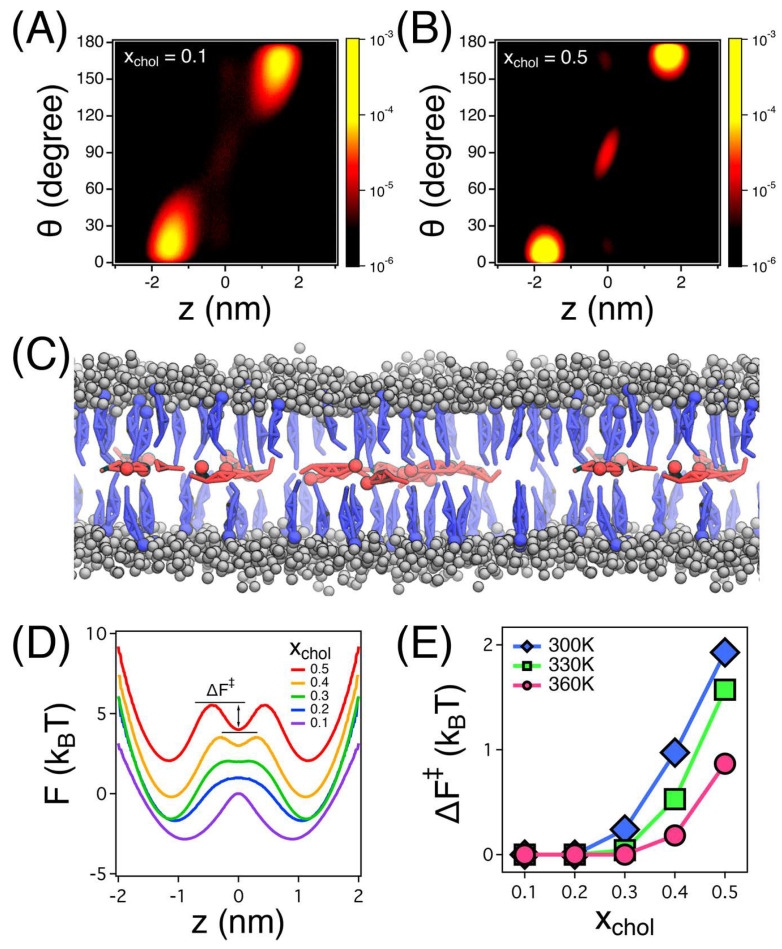
Cholesterol distribution and energy profiles in lipid bilayers. (**A**,**B**) Density distribution functions P(z,θ) of cholesterol molecules at low ($x_{\mathrm{chol}}=0.1$) and high ($x_{\mathrm{chol}}=0.5$) concentrations show a shift from leaflet−localized orientation to increased bilayer center occupancy and reorientation. (**C**) Simulation snapshot at $x_{\mathrm{chol}}=0.5$ illustrates cholesterol localization: molecules in the bilayer center (blue) and within leaflets (red); lipid head groups are shown as gray spheres, and tail groups are omitted for clarity. (**D**) Free energy profiles F(z) at T = 330 K demonstrate the emergence of a metastable state at the bilayer center as cholesterol concentration increases. (**E**) Energy barriers ($\Delta F^{‡}$) for cholesterol transition between center and leaflet positions increase with cholesterol concentration and temperature, indicating reduced cholesterol dynamics at higher concentrations. Together, these panels highlight how increasing cholesterol content alters its spatial distribution, stabilizes its position at the bilayer center, and reduces its diffusional behavior, thereby influencing overall membrane organization and behavior. Reproduced from [67], with permission. Copyright (2018) American Chemical Society.

**Figure 6 membranes-15-00173-f006:**
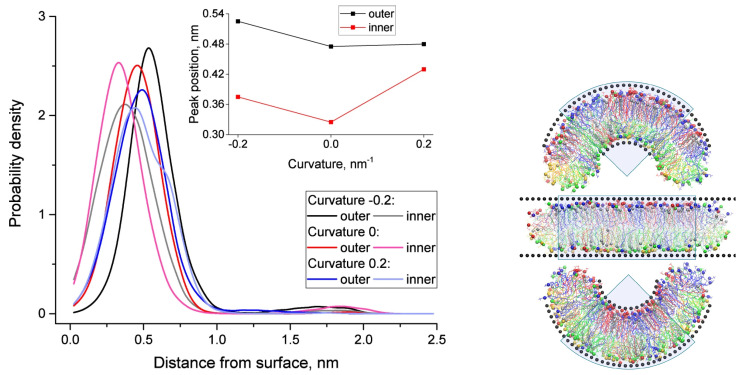
(**Left**) Spatial distribution of cholesterol molecules in inner and outer monolayers under varying membrane curvature. The distance from the membrane surface to the OH group of cholesterol is shown, with the inset highlighting the position of the major peak in cholesterol distribution as a function of curvature. Cholesterol’s distinct spatial arrangements contribute to the stabilization of curved membrane regions. (**Right**) Snapshots of simulated membrane systems with curvature values of c=0.2 nm^−1^ (**top**), c=0 nm^−1^ (**middle**), and c=−0.2 nm^−1^ (**bottom**). Wall particles are shown as black spheres, while lipid types are color-coded: PC (blue), SM (red), PE (green), and PS (yellow). Cholesterol molecules are represented in gray, with head groups depicted as spheres. Shaded sectors indicate regions used for analysis, emphasizing cholesterol’s spatial adaptation to varying curvature conditions. Adopted from [28] under the terms of the Creative Commons CC BY license.

**Figure 7 membranes-15-00173-f007:**
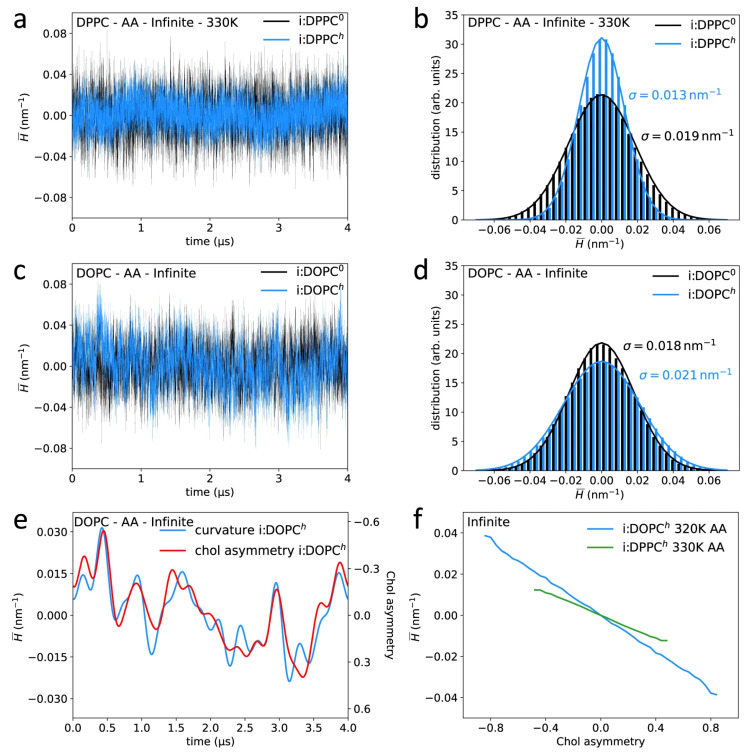
Analysis of cholesterol’s impact on membrane curvature in DPPC and DOPC bilayers. The figure highlights the time-dependent development and distribution of mean curvature (panels (**a**–**d**)), as well as the asymmetry in cholesterol distribution between leaflets (panel (**e**)). Panel (**f**) shows the correlation between mean curvature over time and cholesterol asymmetry; here, cholesterol is responsible for holding bilayers together under different curvatures. All simulations were conducted at 310 K. DPPC systems are shown in red and DOPC in blue; cholesterol-containing systems use solid lines, while cholesterol-free systems are shown with dashed lines. A linear regression fit is applied to the data in panels (**b**,**d**) to indicate the relationship between curvature and cholesterol asymmetry. Reproduced from [84] under the terms of the Creative Commons CC BY license.

**Figure 8 membranes-15-00173-f008:**
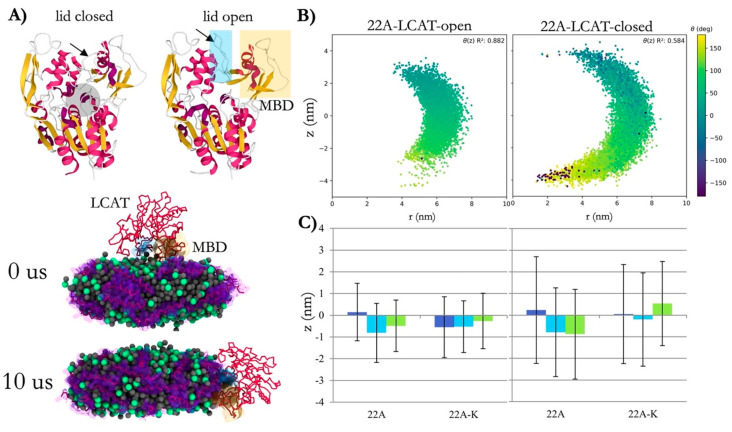
Lecithin–cholesterol acyltransferase (LCAT) structures in nanodiscs. (**A**) X-ray structures of open (6MVD) and closed (4XWG) LCAT highlight the membrane-binding domain (yellow box) and lid (blue box), with the active site indicated by a transparent gray sphere. Simulation snapshots show the dynamic relocation of open LCAT to the nanodisc edge after 10 μs. (**B**) Orientation and *z*-axis positioning of LCAT relative to the nanodisc plane reveal spatial correlations between angle and distance from the nanodisc center. (**C**) Average *z*-axis distances for open and closed LCAT systems (22A and 22A-K) demonstrate distinct spatial distributions, with error bars representing standard deviations. Adapted from [105] under the terms of the Creative Commons CC BY license.

**Figure 9 membranes-15-00173-f009:**
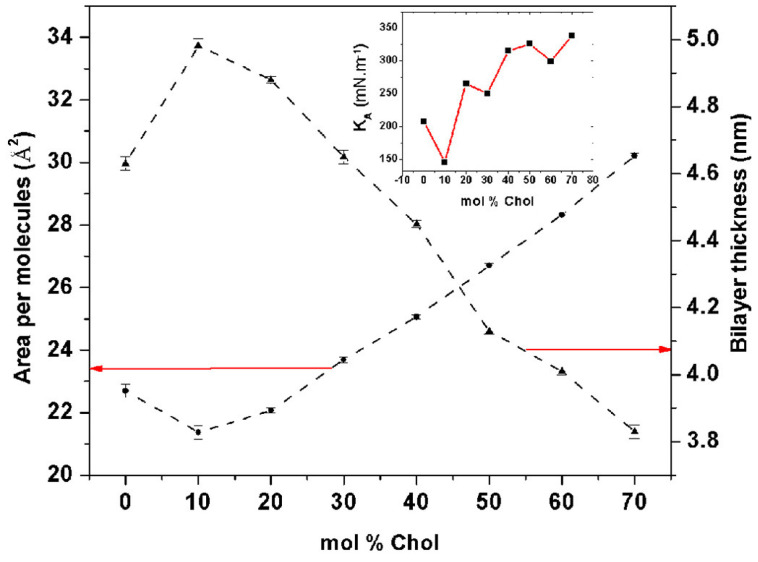
Simulations of niosome bilayer area per molecule ($A_{0}$), bilayer thickness (*d*), and isothermal area compressibility ($K_{A}$) for varying cholesterol levels. The simulation also illustrates that increasing cholesterol content leads to bilayer expansion, with optimal stability observed at 40–50 mol% cholesterol. These results demonstrate that cholesterol modulates the stability and flexibility of niosome bilayers. Reproduced from [2], Copyright (2018), with permission from Elsevier.

**Figure 10 membranes-15-00173-f010:**
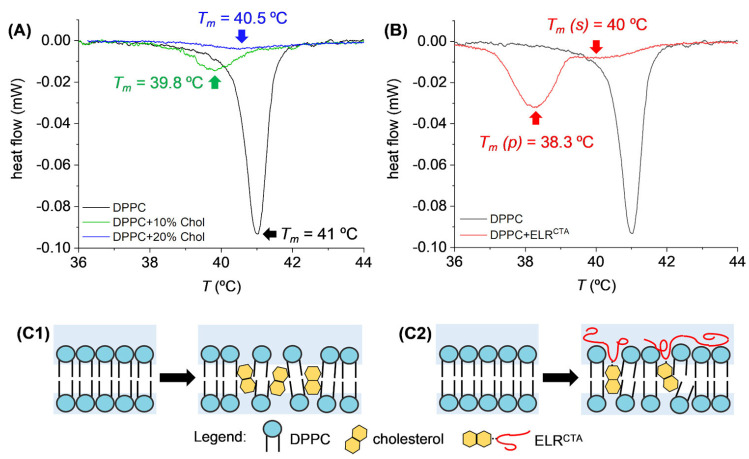
Differential scanning calorimetry (DSC) curves and schematics of DPPC liposomes in the presence of cholesterol and elastin-like recombinamer (ELRCTA). (**A**) Calorimetric response of DPPC liposomes with 10% and 20% cholesterol, highlighting local minima and phase transition temperatures ($T_{m}$). (**B**) Impact of ELRCTA on the phase transition of DPPC. (**C1**) Schematic showing cholesterol intercalating into the non-polar region of the lipid bilayer. (**C2**) Illustration of ELRCTA acting as an anchoring mechanism, with the CTA embedded in the bilayer and the ELR segment exposed in the superficial polar region. The schematics represent lipid bilayers in the gel phase ($T<T_{m}$). Adapted with permission from [115], Copyright (2024), Elsevier.

**Figure 11 membranes-15-00173-f011:**
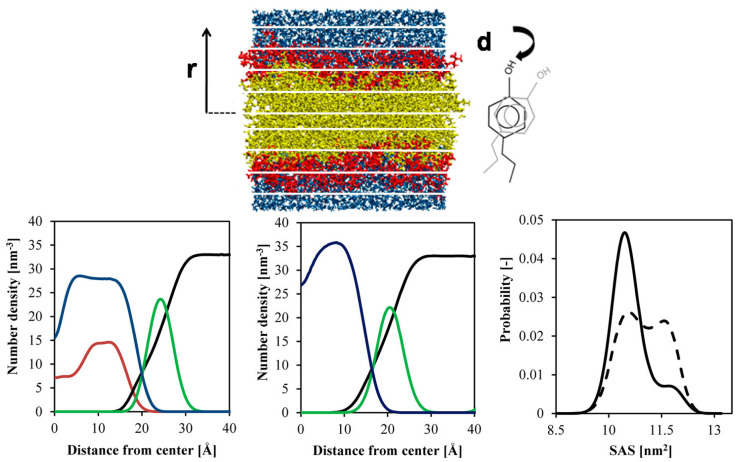
(**Top**) Principle of COSMOmic applied to a lipid bilayer and a solute, illustrating lipid head groups (red), lipid tailgroups (yellow), and water molecules (blue). (**Bottom**) Number density maps for DPPC bilayers of 30% cholesterol (left) and 0% cholesterol (middle): distributions for water (black), DPPC tails (blue), DPPC phosphocholine (green), and cholesterol (red). The right panel shows the probabilistic distribution of the solvent-accessible surface area (SASA) of DPPC conformers in 30% cholesterol (solid line) and 0% cholesterol (dashed line) DPPC bilayers. Adopted from [116], with permission. Copyright (2013) American Chemical Society.

**Figure 12 membranes-15-00173-f012:**
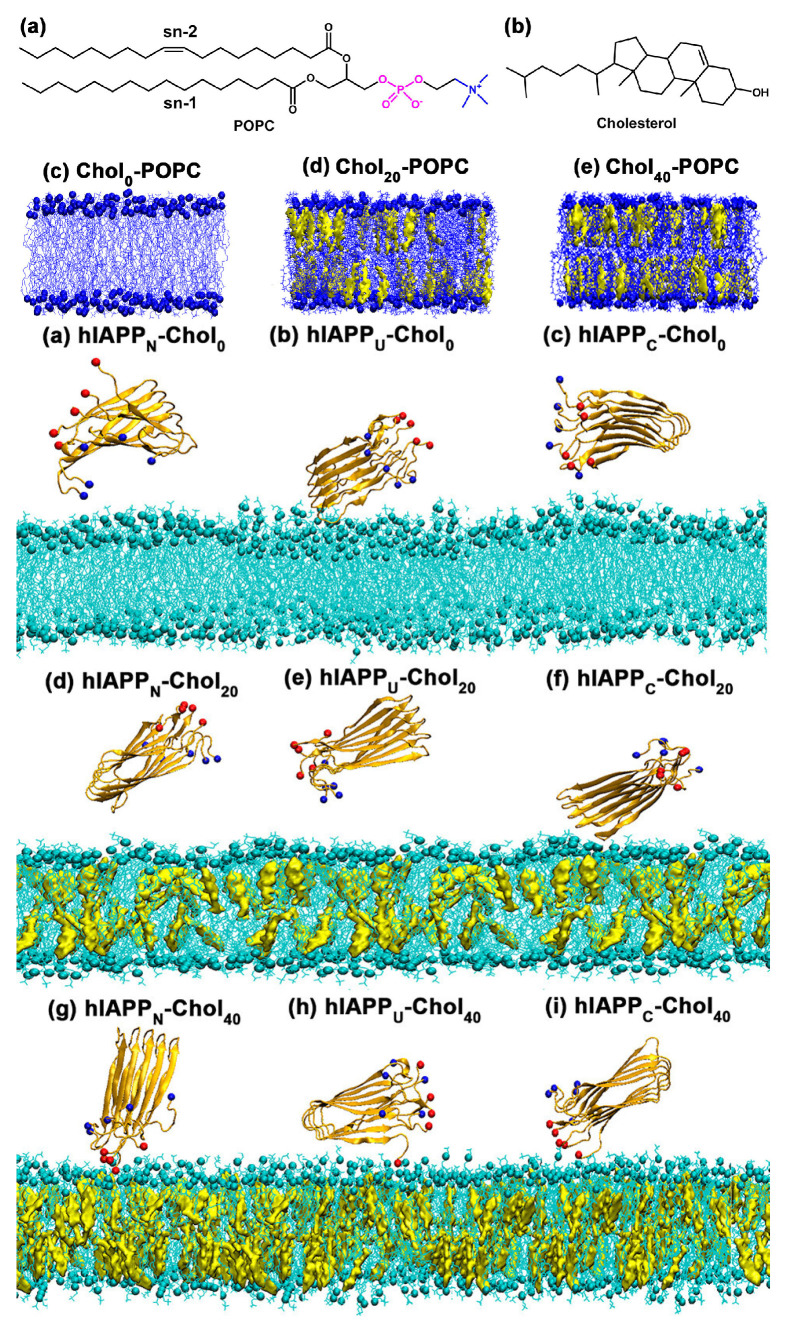
Structural characterization and interaction dynamics of POPC bilayers with varying cholesterol contents. (**Top**) Side views of POPC bilayers containing 0, 20, and 40 mol% cholesterol, highlighting the molecular arrangement of POPC (blue) and cholesterol (yellow) ((**a**) Chemical structure of POPC, (**b**) Chemical structure of cholesterol). (**Bottom**) Final MD simulation snapshots showing the adsorption or desorption of hIAPP pentamers in N-terminal (N), U-shape (U), and C-terminal (C) orientations on POPC bilayers with 0, 20, and 40 mol% cholesterol. Color code: POPC lipid (cyan), cholesterol (yellow), C-termini (red spheres), and N-termini (blue spheres). Adopted from [118], with permission. Copyright (2020) American Chemical Society.

## Data Availability

No new data are reported in this review article.

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
