# Peer review of "Comprehensive Insights into the Cholesterol-Mediated Modulation of Membrane Function Through Molecular Dynamics Simulations"

_membranes, 2025, doi:10.3390/membranes15060173_

Round 1

Reviewer 1 Report

Comments and Suggestions for Authors

This review addresses the multifaceted role of cholesterol in both biological and synthetic membrane systems, with particular emphasis on insights gained from molecular dynamics (MD) simulations. Given cholesterol’s critical functions in modulating membrane fluidity, phase behavior, curvature, and protein interactions, a comprehensive synthesis of simulation data is both timely and relevant. The manuscript has the potential to contribute to the field of membrane biophysics and nanostructure engineering. I can recommend this review, provided the following points are carefully addressed:

  1. Line 36: The abbreviation “li-ordered” is uncommon and may confuse readers. Please consider replacing it with the full term liquid-ordered or the widely accepted acronym Lo.
  2. Lines 72–81: The paragraph repeats previously stated information about membranes and the role of cholesterol, without effectively introducing the “planar bilayers” section. A more focused and distinct introductory paragraph would improve the structure and flow of this section.
  3. Line 75: The sentence lacks logical consistency. The description of inwardly oriented hydrophilic heads contradicts the well-established structural organization of lipid bilayers. Please revise it.

  4. Line 78: A misplaced word in this sentence disrupts comprehension. Please review the sentence structure to ensure clarity.

  5. Lines 199–207: Since the role of cholesterol in lipid membranes is well supported by experimental data, and the primary contribution of this review is to highlight recent simulation-based insights, it is essential that the authors clearly distinguish between experimentally derived and computationally derived results when citing studies, for example, reference 70. Please verify this distinction consistently throughout the manuscript.

  6. Redundancy: Certain points are repeated unnecessarily across the manuscript, particularly in lines 36, 212, and 216, as well as in lines 88 and 169. These repetitions reduce the overall clarity and conciseness of the text. I suggest consolidating or rephrasing to avoid redundancy.

  7. Line 361: The statement regarding cholesterol’s role in liposome structure and function is partially correct and imprecise. Cholesterol is not the sole determinant; it acts as a modulator of membrane properties. 

Author Response

Please see the attached response.

Reviewer 2 Report

Comments and Suggestions for Authors

This review manuscript titled “Comprehensive Insights into the Cholesterol-Mediated Modulation of Membrane Function through Molecular Dynamics Simulations” explores how cholesterol influences biological membranes and their function, with the findings from the molecular dynamics simulation studies. This review manuscript is informative and provides valuable insights of cholesterol in a structured manner. However, the manuscript would benefit from a few minor corrections. Addressing these issues would enhance the manuscript's readability, impact, and broaden its relevance.

Comment 1.

The authors claim that this review is comprehensive. However, there is a lack of information regarding certain important aspects, e.g., cholesterol-cyclodextrin interactions, which are used in the experimental settings to modulate cholesterol content and have huge implications for membrane function and drug delivery, topics widely discussed in this manuscript. Including a discussion on this topic (studied with molecular dynamics simulation) would enhance the review article.

Comment 2.

Abstract. The phrase “adding rigidity and fluidity” is counterintuitive and potentially confusing. A more detailed mechanistic explanation is necessary to clarify how cholesterol can contribute to both properties under different conditions.

Comment 3.

Figure 1. The phrase "along the membrane thickness (z-axis)" could cause confusion, particularly in a 2D plot. If the use of the "z-axis" is intentional, it would be helpful to clarify that the z-axis represents the depth or thickness of the membrane in this context. A more explicit explanation would improve clarity.

Comment 4.

In the figure caption, it is written: “Analysis of cholesterol’s impact on membrane curvature in DPPC and DOPC bilayers.” However, it is unclear which colors in the figure correspond to systems with or without cholesterol. Moreover, the figure caption lacks some key elements: the temperatures used and the type of fit applied in panels B and D. These details would improve the clarity of the provided figure.

Comment 5.

All figure citations in the text should be formatted consistently without the use of brackets.

Comment 6. Starting from line 390, the sentence “Findings revealed a dual effect of cholesterol… lower curvature and reduced drug release” describes the effect of lipids, rather than cholesterol, on drug release efficiency. However, the concluding remark states, “Such results highlight the role of cholesterol and lipid types like MSPC in influencing liposome behaviour for best-fit drug delivery systems.” Please provide a more direct explanation of cholesterol’s effect on liposome behaviour related to drug release.

Other comments

Line 26: “between Li-ordered (Lo) and Li-disordered (Ld) domains” is unclear and confusing. Please clarify the sentence (liquid-ordered…).

Line 78: “and resistance to stresses citerisselada2019cholesterol”, there seems to be a citation error in the text. Please correct the sentence.

Line 114. The abbreviation “SCD” should be written in uppercase letters to maintain consistency. Also, some abbreviations used throughout the manuscript are not present in the abbreviation list.

Line 165. The term “speedy diffusion” is not recommended for scientific language. More precise terms like rapid, fast, or high diffusion rate would be more appropriate.

Line 212: Please correct the typing “Lio”

Line 220: The phrase “dynamical dynamics” is redundant. Please revise it to avoid repetition, such as dynamics. Also, in the same paragraph, there is an error in the sentence: “environments is central to understanding things,". Since “environments” is plural, it should be followed by “are”.

Lines 226-227: The term "protein classification" is unclear and inappropriate in this context. Protein classification typically refers to the categorization of proteins based on structure or function. Please revise the term to a more appropriate term.

Line 320: Please explain to the readers the abbreviation “HDL”.

Comments on the Quality of English Language

There are grammar errors throughout the manuscript, which could be corrected. 

Author Response

Please see the attached response.

Reviewer 3 Report

Comments and Suggestions for Authors

This manuscript titled "Comprehensive Insights into the Cholesterol-Mediated Modulation of Membrane Function through Molecular Dynamics Simulations" presents a thorough and well-structured review of the effects of cholesterol on various membrane systems, including planar bilayers, curved membranes, liposomes, nanodiscs, and proteoliposomes. The authors successfully consolidate findings from recent molecular dynamics (MD) simulations to elucidate the multifaceted roles of cholesterol in modulating membrane properties. It requires major revision before publication:

  1. The introduction outlines cholesterol’s importance in membrane biology but does not clearly define the review’s scope, specific objectives, or how it differs from existing reviews. The authors should briefly state the key novelty and scientific question this review aims to address.
  2. Although the review cites extensive literature, some sections lack deeper mechanistic insight or comparative analysis. For instance, in the phase separation subsection, the authors could better distinguish cholesterol's behavior in saturated vs. unsaturated lipid systems. It would also strengthen the review to compare cholesterol with similar sterols such as ergosterol or phytosterols, especially in the context of membrane ordering and fluidity.
  3. Key concepts such as cholesterol-induced membrane thickening and lipid ordering are discussed multiple times (e.g., in the abstract, introduction, and planar bilayer section) with minimal additional information. The manuscript would benefit from consolidating overlapping content to avoid redundancy and improve flow.
  4. Figures are comprehensive but often contain dense multi-panel data with limited explanation in the captions. For example, Figure 5 includes multiple panels with technical plots, but the caption does not clearly summarize the key message. Simplifying figure layouts and enhancing captions with interpretative comments would improve clarity.
  5. The conclusion summarizes the content effectively but lacks specific, actionable suggestions for future research. The authors could recommend, for example, the application of hybrid MD methods (e.g., QM/MM or enhanced sampling) or outline key open questions in cholesterol-membrane interactions under physiological stress.

Author Response

Please see the attached response.

Round 2

Reviewer 3 Report

Comments and Suggestions for Authors

The author has addressed all my concerns, and the manuscript is now ready for publication.